# Characterization of Incidental Pathogenic Germline Findings Detected via ctDNA among Patients with Non-Small Cell Lung Cancer in a Predominantly Hispanic/Latinx Population

**DOI:** 10.3390/cancers16061150

**Published:** 2024-03-14

**Authors:** Esha Vallabhaneni, Samuel A. Kareff, Reagan M. Barnett, Leylah M. Drusbosky, Shivani Dalal, Luis E. Raez, Edgardo S. Santos, Federico Albrecht, Mike Cusnir, Estelamari Rodriguez

**Affiliations:** 1Mount Sinai Comprehensive Cancer Center, Miami Beach, FL 33140, USA; esha.vallabhaneni@msmc.com (E.V.); mike.cusnir@msmc.com (M.C.); 2University of Miami Sylvester Comprehensive Cancer Center, Jackson Memorial Hospital, Miami, FL 33136, USA; 3Guardant Health, Redwood City, CA 94063, USA; rbarnett@guardanthealth.com (R.M.B.);; 4Memorial Cancer Institute, Pembroke Pines, FL 33028, USA; sdalal@mhs.net (S.D.); lraez@mhs.net (L.E.R.); 5Charles E. Schmidt College of Medicine, Florida Atlantic University, Boca Raton, FL 33431, USA; edgardo_ny@hotmail.com; 6Miami Cancer Institute, Miami, FL 33176, USA; dralbrecht@baptisthealth.net

**Keywords:** non-small cell lung cancer, ctDNA, pathogenic germline variants

## Abstract

**Simple Summary:**

Our research demonstrates that genetic factors may play a significant role in the development of non-small cell lung cancer (NSCLC). We focused on the Hispanic/Latinx community in South Florida to determine if certain germline (or inherited) variants in DNA were expressed in patients diagnosed with NSCLC. We found that certain germline variants were present at a higher rate in the Hispanic/Latinx community. Identification of these variants may change the management of their cancer.

**Abstract:**

Pathogenic germline variants (PGVs) may be under-detected as causative etiologies in patients with non-small cell lung cancer (NSCLC). The prevalence of PGVs has been reported between 1 and 15% of patients, depending on the patient population. The rate within Hispanic/Latinx populations remains unknown. We retrospectively analyzed the genomic results (Guardant360, Redwood City, CA, USA) of 878 patients with advanced or metastatic NSCLC at five centers in South Florida, USA, from 2019 to 2022 to analyze the rate of incidental PGVs (iPGVs) identified via circulating cell-free tumor DNA (ctDNA). We then stratified the results by tumor histology, age, gender, race, ethnicity, genetic pathway, and co-mutations. Twenty-one iPGVs were identified (21/878 = 2.4%). Among the 21 iPGVs identified, 14 patients were female (66.7%) and 7 were male (33.3%), with a median age of 67 years and tobacco history of 2.5 pack-years. In total, 52.4% of patients identified as Hispanic/Latinx (*n* = 11) of any race; 19.0% as Ashkenazi Jewish (*n* = 4), 9.5% as non-Hispanic/Latinx black (*n* = 2), and 19.0% as non-Hispanic/Latinx white (*n* = 4). iPGVs in the homologous recombination repair pathway were solely expressed in this cohort (10 *ATM*, 8 *BRCA2*, and 3 *BRCA1*). In total, 76% (16/21) of patients with iPGVs co-expressed somatic alterations, with 56% (9/16) demonstrating alterations in targetable genes. Overall, our real-world findings offer a point prevalence of iPGVs in patients with NSCLC of diverse populations, such as patients who report Hispanic/Latinx ethnicity.

## 1. Introduction

Lung cancer arises from both somatic and heritable factors [1]. While lung cancer heritability has been estimated at 18%, there is little information regarding the genetic basis of lung cancer within Hispanic/Latinx populations [2]. Emerging data highlight the possible existence of additional, previously undescribed pathogenic germline variants (PGVs) as causative etiologies in non-small cell lung cancer (NSCLC). PGVs in cancer susceptibility genes may lead to predisposition in tumorigenesis. Clinically, tumor next-generation sequencing (NGS) circulating tumor DNA (ctDNA) assays are often utilized to examine NSCLC for the presence of actionable somatic alterations [3], but these assays can also identify incidental PGV (iPGV) findings to be confirmed by validated germline laboratories.

Expanding data demonstrate that patients harboring certain PGVs are associated with an increased risk of developing lung cancer. Two major syndromes that are contributors to hereditary lung cancer involve variants of *TP53* and *EGFR*, but additional genes that have been implicated include *ATM*, *BRCA1*, *BRCA2*, *YAP1*, *HER2*, and *CHEK2*, among others [4]. Patients with a germline *TP53* mutation, or Li-Fraumeni Syndrome, have a 2.3–6.8% chance of lung cancer development [5], with the greatest frequency occurring in lung adenocarcinoma [6]. In addition to *EGFR* T790M mutations being associated with familial clustering of lung cancer [7,8], this variant has also been linked to non-tobacco users [9,10].

While lung cancers primarily arise from somatic alterations, a distinct subset can be influenced by PGVs that interact with the environment to shape the tumor landscape [11]. The dependence of tumors on PGVs is variable and often dictated by both penetrance and lineage [12]. Germline *EGFR* T790M mutations are the most commonly reported PGVs in lung cancer, specifically in NSCLC. A recent study demonstrated that germline prevalence of *EGFR* T790M is consistent with Mendelian inheritance and, furthermore, that the carriers of the mutation were shown to have an increased rate of lung nodules without a cancer diagnosis, and half of the carriers were diagnosed with cancer by the age of 60 [13]. The overall prevalence of PGVs may vary between 1 and 15% of patients, depending on the patient population studied [14]. Genetic testing for moderate penetrance cancer susceptibility genes associated with a greater than two-fold increased risk of cancer is now commonplace, and guidelines incorporate utilizing screening for these genes for prevention [15]. Germline testing may also guide an individual’s reproductive decisions, in the setting of inherited cancer predisposition syndromes, for family planning purposes [16]. Of note, the rate of PGVs may be underestimated by somatic testing platforms, with one retrospective study demonstrating that 8.1% of PGVs that were revealed during germline testing were missed in patients who had already undergone prior tumor DNA sequencing [17].

Additionally, the rate of PGVs within Hispanic/Latinx populations remains unclear. In Latin American-born Hispanics undergoing genetic testing for hereditary breast or ovarian cancer, the overall yield of PGVs was significantly higher than that detected in U.S.-born Hispanics [18]. Variations in ancestry and/or ethnicity among Latin American populations also lead to varying rates of PGV penetrance. For example, in gastric cancer, studies found that an increased risk of developing cancer is associated with the different germline risk variants based on which region of Latin America the population hails [19].

Furthermore, analyses of *EGFR* mutations in the Hispanic population within Latin America have shown that rates vary between the countries. This variation may be due to the citizens’ ancestries. For example, *EGFR* mutations are the lowest in Argentina and Uruguay where there is a strong history of European ancestry, and highest in Peru where the majority of the population is of indigenous descent [20]. This implies that germline risk variants such as PGVs could be the link between ethnicity and risk for *EGFR*-mutant lung cancer.

Therefore, the aim of this study was to identify iPGVs, as well as any secondary alterations in targetable genes, within the racially and ethnically diverse community of South Florida, which is predominantly of Hispanic/Latinx ethnicity. Knowledge of the relationship between both primary and incidental PGVs within the Hispanic/Latinx community is currently lacking, and further understanding might be applied to precision oncology to improve the management of this patient population.

## 2. Materials and Methods

We retrospectively analyzed the genomic results (Guardant360, Redwood City, CA, USA) performed as part of routine clinical care of 878 patients with advanced or metastatic NSCLC at five academic or community centers in South Florida, USA, from 2019 to 2022 to assess the rate of iPGVs identified via ctDNA. Guardant360 is validated to detect somatic variants in circulating cell-free DNA (cfDNA) and is not intended to identify or diagnose a hereditary condition, but incidental germline findings in a subset of genes may be commented on. Validated germline tests utilizing buffy coat or other sample materials validated for assessing germline findings may identify a higher frequency of germline alterations in a similar population. All samples were analyzed for somatic alterations in up to 83 genes (including insertions/deletions in *EGFR* exons 19/20) and germline alterations in 3 genes (*ATM*, *BRCA1*, *BRCA2*). After centrifugation of whole blood, 5–30 ng of cell-free DNA isolated from plasma was processed for digital next-generation sequencing (NGS), in accordance with its previously described validation [21,22]. iPGVs are reported for mutations with a variant allele frequency suspicious for germline origin (e.g., based on variant allele fraction cut-offs and in-sample algorithms) and annotated as suspicious for germline origin per expert consensus. Most variant allele fractions suggestive of germline alterations are >30%. Germline mutations were annotated using a proprietary bioinformatic algorithm previously described and validated [21,23,24,25]. These results were then stratified by tumor histology, age, gender, race, ethnicity, and co-mutations. Ethnicity and race were self-reported by patients and collected via retrospective chart review. Statistical testing was performed using Fisher’s Exact *t*-tests using GraphPad Prism (Boston, MA, USA).

## 3. Results

Twenty-one (2.4%) patients had an iPGV identified via ctDNA testing during the study period. The median age at time of detection was 67 years. Fourteen patients (66.7%) were female, and seven patients (33.3%) were male. Tobacco history was available for 19 of 21 patients: median 2.5 pack-years, range 0–60 pack-years (interquartile range 40 pack-years). The majority of iPGVs were identified in patients of white race (19/21 = 90.48%). The remainder of iPGVs (2/21 = 9.52%) were identified in patients of black race.

A total of 433 patients of Hispanic/Latinx ethnicity were included in the full cohort, and 11 iPGVs were identified in patients of Hispanic/Latinx ethnicity (11/21 = 52.38%), which represented 2.54% of the Hispanic/Latinx population (11/433) included in our analysis. Four iPGVs were observed in patients with reported Ashkenazi Jewish ethnicity (4/21 = 19.05%). Finally, there were six iPGVs noted in patients that did not report Hispanic/Latinx or Ashkenazi Jewish ethnicity, and there were no iPGVs in patients identifying as both Ashkenazi Jewish and Hispanic/Latinx (Table 1).

The prevalence of iPGVs by racial and ethnic sub-groups was next analyzed. According to racial sub-groups, the white population harbored significantly more iPGVs than the black population (*p* < 0.0001). Regarding ethnic sub-groups, Hispanic/Latinx patients demonstrated significantly more iPGVs than patients identifying as both Hispanic/Latinx and Ashkenazi Jewish (*p* < 0.001) or with unknown ethnicity (*p* < 0.001). A notable trend in differences in iPGV prevalence between patients identifying as Hispanic/Latinx and those identifying as Ashkenazi Jewish (*p* = 0.052) was detected. There was no difference in prevalence between patients identifying as Hispanic/Latinx compared to those with neither Hispanic/Latinx or Ashkenazi Jewish ethnicity (*p* = 0.21) (Figure 1).

Of the 21 patients with iPGVs, 16 patients also had co-occurring somatic alterations detected (16/21 = 76%; Figure 2). Among these 16 patients with an iPGV and at least one somatic alteration identified, eleven patients identified as female (68.8%) and five patients as male (31.2%), with a median age of 65.5 years. Overall, 50% of patients identified as Hispanic/Latinx (*n* = 8) of any race; 25% as Ashkenazi Jewish (*n* = 4); 6.2% as non-Hispanic/Latinx Black (*n* = 1); and 18.8% as non-Hispanic/Latinx White (*n* = 3). iPGVs in the homologous recombination repair (HRR) pathway were solely represented in this cohort, with ten ATM, eight BRCA2, and three BRCA1 variants identified. There were nine patients with iPGVs (56.2%) who had at least one alteration in an actionable gene identified (EGFR, BRAF, KRAS; Table 2).

## 4. Discussion

This study assessed the prevalence of iPGVs in NSCLC within the predominantly Hispanic/Latinx population in South Florida, U.S. These data demonstrate that germline alterations in the homologous recombination repair pathway (namely, *ATM*, *BRCA2*, and *BRCA1*) among Hispanic/Latinx patients with NSCLC are not uncommon at 2.54%, a rate nearly double that observed in the non-Hispanic/Latinx and non-Ashkenazi Jewish populations represented in our cohort.

A population’s ethnic and ancestral backgrounds influence the prevalence of specific PGVs. For example, people with Ashkenazi Jewish ancestry demonstrate a higher frequency of PGVs compared to those of non-Ashkenazi Jewish ancestry, and a higher proportion of patients with Ashkenazi ancestry were carriers of moderate to high penetrance variants when compared to patients of other ancestries [12]. This trend was also observed in our findings, as 6.35% (4/63) of the total patients identifying as Ashkenazi Jewish ethnicity demonstrated iPGVs.

Other populations have been demonstrated to harbor an increased prevalence of specific PGVs. Additional examples include populations of African ancestry in which there is a potential association of *BRCA2* in squamous NSCLC [26]. Conversely, *EGFR* T790M germline mutations are reported with decreased prevalence in East Asian populations in comparison to North American populations, despite it being the most frequently reported PGV in lung cancer [27]. Variance in rate is also observed based on type of cancer; for example, in Hispanic patients with prostate cancer, germline variants were identified in 3.8% of patients [28], and between 9.1% and 18.7% were identified in Hispanic patients with hereditary breast and/or ovarian cancer depending on their area of origin [18].

A significantly higher proportion of high-penetrance PGVs predisposing patients to lung cancers, including *TP53*, *EGFR*, and *BAP1*, have been detected in patients who had multiple primary tumors, family history of any cancer, or early age of diagnosis compared to noncarriers [29]. In addition, patients with a family history of lung cancer have been shown to have an increased incidence of cancer regardless of tobacco use, indicating that genetic characteristics increase the likelihood of smokers and non-smokers alike developing lung cancer [30,31,32].

Germline variants in *TP53*, causing Li-Fraumeni syndrome, are known to contribute to increased risk of development of cancers such as breast cancer or sarcoma [33]. However, developing information indicates that germline mutations in the *TP53* gene may also contribute to increased development of NSCLC, in particular, *EGFR*-mutant NSCLC [34]. In a study that observed the relationship between patients with germline *TP53* variants and NSCLC, driver *EGFR* mutations were seen in the majority of the cases, with exon 19 deletion being the most common alteration [35]. Another study found that, in particular, the germline variant *TP53* p.R337H was most frequently associated with the development of *EGFR* somatic mutations [36]. This may indicate that a history of lung cancer in first-degree relatives in this subset of patients with germline *TP53* mutations may mean that they have an increased risk of lung cancer regardless of environmental factors.

In East Asian populations, the odds ratio of NSCLC development in patients with known PGVs was significantly higher when compared to those without, and pathogenic mutations were most commonly observed in *BRCA2*, followed by *CHEK2* and *ATM* [37]. Frameshift and nonsense mutations were the leading types of mutations seen in a study that examined germline *BRCA*1 and *BRCA*2 mutations in a Chinese population with NSCLC [38].

Studies from The Cancer Genome Atlas (TCGA) have demonstrated that, within NSCLC, approximately 5.4% of squamous cell lung cancers and 6.4% of lung adenocarcinomas harbor pathogenic or likely pathogenic variants [1,39,40]. Additionally, these studies demonstrate that germline variants in *ATM* are one of the most frequent germline alterations in lung adenocarcinoma. Similarly, germline *ATM* variants have been strongly associated with moderate penetrance in catalyzing lung adenocarcinoma [41]. One study that examined PGVs in lung adenocarcinoma found them to occur at a comparable or slightly lower rate as TCGA at 2.5–4.5%, and also found that mutations most commonly occurred in *ATM* (50%), followed by *TP53* (28.6%), *BRCA2*, *EGFR*, and *PARK2*, all (7.1%) [42]. However, another study that compared the rate of prevalence of *BRCA1* and *BRCA2* PGVs in patients with lung adenocarcinoma with *ATM* PGVs reported a higher prevalence of *BRCA1* and *BRCA2* PGVs [29]. While *ATM* germline variant testing was not previously recommended by the European Society for Medical Oncology (ESMO) Precision Medicine Working Group as a common inclusion for clinical germline testing, it is now recommended for patients undergoing testing for actionable pathogenic variants in addition to seven ‘most actionable’ cancer-susceptibility genes (*BRCA1*, *BRCA2*, *PALB2*, *MLH1*, *MSH2*, *MSH6*, and *RET*) for which germline investigation is recommended—regardless of tumor type [43,44].

Knowledge of *BRCA1* and *BRCA2* status in NSCLC could play a role in medical management based on the clinical rationale that *BRCA*-altered tumors are more sensitive to platinum salts [45]. Additionally, poly-ADP-ribose (PARP) inhibitors have been reported to demonstrate clinical benefit in germline *BRCA*-associated cancer types, irrespective of tumor origin [46]. While PARP inhibitor therapy is not currently standard-of-care treatment in the management of NSCLC, its use has been shown to potentially improve progression-free survival in NSCLC; larger trials are required to establish treatment effects, especially for patients with NSCLC and *BRCA* mutations [47].

The role of *BRCA1/2* in NSCLC remains controversial, with some studies demonstrating that the variants are associated with an increased risk for lung cancer, while others do not [48]. In one study, *BRCA2* PGVs were found in 0.8% of cases of NSCLC, which is higher than the estimated frequency in the general East Asian population [49]. However, other studies have found the rate of *BRCA2* PGVs in patients with NSCLC to be comparable to control groups without NSCLC [24]. *BRCA2* may be associated with adenocarcinoma given that eight *BRCA2* PGV carriers with NSCLC in one study had this subtype of NSCLC [50], and that in the over 300 patients with NSCLC studied in the SAFIR02-lung trial, the only two identified *BRCA2* PGV carriers had adenocarcinoma [46]. Not withstanding, larger studies need be performed to confirm this association [4]. Utilizing ctDNA to investigate iPGVs is not without potential challenges. For example, when testing for high-penetrance PGVs, unexpected positive results may result in a clinical conundrum in patients whose history and presentation do not suggest inherited cancer syndrome [15]. Higher rates of testing and incorporation of larger multigene panel testing may also be associated with the identification of variants of unknown significance.

Furthermore, due to a lack of current standardization for germline variant classification, there may be discrepancies in classifications between laboratories [51]. Through whole-genome sequencing, PGVs or likely PGVs can be uncovered that are unrelated to the primary reason for testing. However, these incidental PGVs should be disclosed to patients [15]. Additionally, ethnic and racial disparities are not only identified in terms of cancer mortality but are also prevalent in genetic testing and subsequent management with screening and risk reduction [15]. Addressing these issues will allow for more robust and accurate genomic profiling via ctDNA and more comprehensive patient care.

Despite these challenges, the use of ctDNA in clinical practice has many beneficial outcomes that are still expanding. In some situations, ctDNA testing can yield greater information than tissue-based genotyping, and it can also be used in situations where tissue is challenging to obtain [52,53]. Furthermore, ctDNA can be useful in patients with metastatic disease due to tumor heterogeneity [54]. Given the short half-life of ctDNA, disease monitoring can occur in real-time [55]; however, its role in treatment surveillance and screening is still being determined, especially when addressing the role of iPGVs.

*Study Limitations*. Our study is not without limitations. While the ctDNA assay used can detect incidental germline findings, it is not a clinical germline assay and therefore has not been validated for germline testing. As such, these results should be reconciled with NCCN guideline-based recommendations for *ATM*, *BRCA1*, and *BRCA2* PGVs. Furthermore, the assay used in our study only analyzes a specific subset of genes for potential germline events, and there may be additional altered genes that were not evaluated in this cohort. Additionally, this South Florida cohort is representative of a small geographical location and may not correspond to results seen in other Hispanic/Latinx populations in the United States or internationally. Our study focuses on patients with a diagnosis of NSCLC, and therefore the true incidence of PGVs in this population may be higher for patients seeking testing for other primary malignancies such as breast, pancreatic, and/or ovarian cancers.

Some challenging aspects of this study included the use of real-world data to generate real-world evidence, which has limitations when compared to dedicated interventional studies [56]. While real-world data and evidence have many potential benefits and implications when applied in oncology care, data collection and the reliability of data with issues such as duplication or missingness remain a challenge [56]. Similarly, given that our study was based on real-world data, accuracy and access to electronic health record data was another demanding aspect of the study. Additionally, some providers lost access to data due to professional changes, which resulted in some incomplete data. Furthermore, limitations of the assay must be taken into consideration.

Despite these limitations, the data herein provide meaningful information on the role of ctDNA in identifying potential germline alterations across ethnicities, especially in Hispanic/Latinx populations.

## 5. Conclusions

Within the Hispanic/Latinx community with advanced or metastatic NSCLC in South Florida, incidental PGVs with alterations in targetable genes were detected via ctDNA testing. Larger studies investigating PGVs in Hispanic/Latinx populations may be impactful for improving screening and management of this population. Moreover, targeting PGVs when providing care may be especially beneficial to the Hispanic/Latinx community of South Florida, and potentially other geographic regions. In Miami-Dade County, this community makes up around 72% of the population and encompasses a large cohort of immigrants from diverse regions such as the Caribbean, Central America, and South America, and includes patients of Hispanic/Latinx ethnicity born in the United States [57]. When compared with their counterparts nationally, the South Florida Hispanic/Latinx population is known to have higher cancer mortality rates [58].

Our study confirms that liquid biopsy platforms such as ctDNA may result in the detection of iPGVs. Previous studies have demonstrated that the combination of tumor sequencing and germline testing increases the chance of detecting clinically relevant changes [59]. Given the increased utilization of ctDNA testing as a standard-of-care assay, this may also improve identification of iPGVs, especially in patients with a family history that is unremarkable for cancer or in populations less likely to seek or be offered genetic testing. Our study demonstrates that, while rare, iPGVs may be identified via ctDNA at rates higher than previously believed.

Overall, more studies with orthogonal germline data are needed to clarify the role of ctDNA in identifying iPGVs, including in specific ancestries. Obtaining germline data may lead to the development of improved genetic screening strategies and management options for patients. Considerations should be made for barriers to precision care for patients in the Hispanic/Latinx community, who may experience differences in access to care, health insurance, and immigration status [60]. Finally, proactive surveillance may be beneficial for the family of patients with NSCLC harboring PGVs in cancer-predisposing genes. This hypothesis could be validated for future prospective studies.

## Figures and Tables

**Figure 1 cancers-16-01150-f001:**
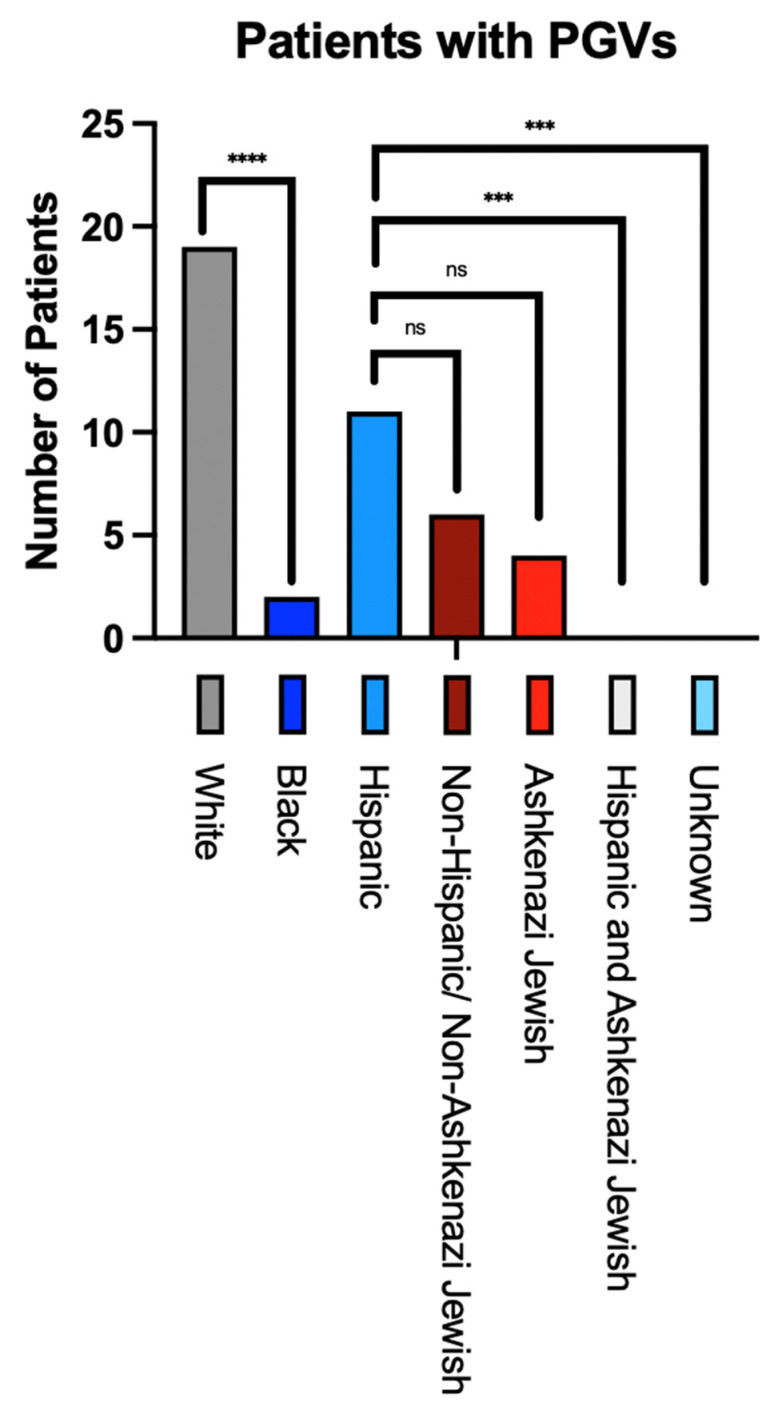
Number of patients with iPGVs, and statistical significance of detecting iPGVs according to predominant racial or ethnic sub-groups within our cohort. This Figure demonstrates the absolute number of patients per demographic group with iPGVs detected in our cohort and compares the statistical likelihood that variations in prevalence exist among racial or ethnic sub-groups. **** = statistically significant racial finding; *** = statistically significant ethnic finding; ns = not statistically significant.

**Figure 2 cancers-16-01150-f002:**
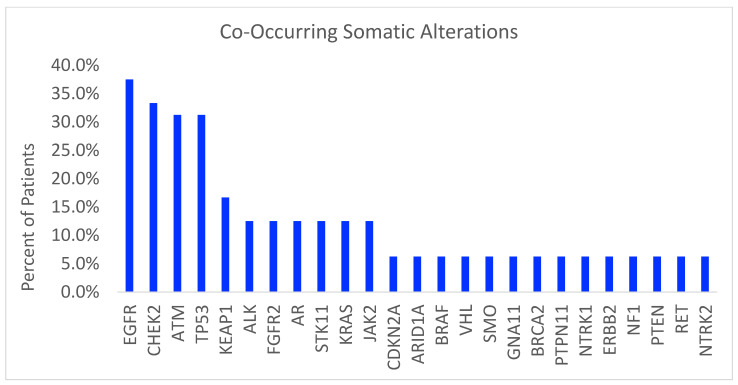
Co-occurring somatic alterations in patients with iPGVs (*n* = 16). This figure depicts as a bar chart the percentage of patients demonstrating specific co-occurring somatic alterations within the patients with iPGVs detected by ctDNA in our cohort.

**Table 1 cancers-16-01150-t001:** Cohort demographics by race and ethnicity with associated iPGV prevalence.

	Total (%)	iPGVs (%)
** *Race* **		
American Indian/Alaska Native	1 (0.11%)	0
Asian	21 (2.39%)	0
Black	79 (9.00%)	2 (9.52%)
Multiple	5 (0.60%)	0
White	759 (86.45%)	19 (90.48%)
Unknown	11 (1.25%)	0
Other	2 (0.23%)	0
** *Ethnicity* **		
Hispanic/Latinx	433 (49.32%)	11 (52.38%)
Ashkenazi Jewish	63 (7.18%)	4 (19.05%)
Hispanic/Latinx and Ashkenazi Jewish	4 (0.46%)	0
Not Hispanic/Latinx or Ashkenazi Jewish	364 (41.46%)	6 (28.57%)
Unknown	14 (1.59%)	0
**Total**	878 (100%)	21 (100%)

This table lists by row racial and ethnic categories and by column the total number of patients and iPGVs detected. Note: sums may not equal 100% due to rounding.

**Table 2 cancers-16-01150-t002:** Total iPGVs and alterations in actionable genes detected in our cohort.

Total iPGVs	*n* = 21 (%)
*ATM*	10 (47.6%)
A1299fs	1 (10%)
E1751fs	1 (10%)
K468fs	1 (10%)
R1882	1 (10%)
R2832C	1 (10%)
R3008C	1 (10%)
Q1970	1 (10%)
Splice Site SNV	3 (30%)
*BRCA2*	8 (38.1%)
E1308	1 (12.5%)
K1530	1 (12.5%)
N257fs	1 (12.5%)
Q2042	1 (12.5%)
Q2943fs	1 (12.5%)
S142I	1 (12.5%)
S1882	1 (12.5%)
T1707fs	1 (12.5%)
*BRCA1*	3 (14.3%)
K894fs	1 (33.3%)
L392fs	1 (33.3%)
T276fs	1 (33.3%)
Total Alterations in Targetable Genes	9 (24.9% total)
*EGFR*	6 (28.6% total)
*EGFR* L858	1 (16.7%)
*EGFR* exon 19 deletion	1 (16.7%)
*EGFR* exon 20 insertion	1 (16.7%)
*EGFR* other	3 (50%)
*BRAF* V600E	1 (4.8% total)
*BRAF* V600E	1 (100%)
*KRAS*	2 (9.5% total)
*KRAS* G12V	1 (50%)
*KRAS* L19F	1 (50%)

This table lists by row the number of iPGVs or total alterations in targetable genes within our cohort in the first column. Totals and corresponding percentages are listed in the second column.

## Data Availability

The data presented in this study are available on request from the corresponding author.

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
