# Peer review of "Characterization of Incidental Pathogenic Germline Findings Detected via ctDNA among Patients with Non-Small Cell Lung Cancer in a Predominantly Hispanic/Latinx Population"

_cancers, 2024, doi:10.3390/cancers16061150_

Round 1

Reviewer 1 Report

Comments and Suggestions for Authors

The manuscript focuses on Characterization of Incidental Pathogenic Germline Findings Detected via ctDNA among Patients with Non-Small Cell Lung Cancer in a Predominantly Hispanic/Latinx Population represents a technically correct manuscript available for the publication on this journal afetr moderate suggestions.

- In the introduction section, please, could the authors identify the potential implications of clinical practice of germline alterations via cfDNA based molecular testing?

- In the methodological section, please, could the authors consider how a germlnen approach based on molecular profiling of other biological source of nucleic acids (buffy coat, normal tissue) may integrate molecular findings identified in this series of lung cancer patients?

- In the methodological section, please, could the authors highlight exon 19-20 EGFR mutations? In my opinion, this point may improve the readability of the manuscript

- In the methodological section, please, could the authors describe in details technical alghoritm adopted to identify germline mutations? In my opinion, this point requires a technical revision

- In the discussion section, please, could the authors overview the most challenging aspects of this study?

Comments on the Quality of English Language

Minor english editing

Reviewer 2 Report

Comments and Suggestions for Authors

The paper is potentially interesting but misleading in the current form.

Please provide concise review on hereditary predisposition for lung cancer.

Germline variants in TP53 and EGFR genes should be acknowledged.

ATM renders only moderate increase in cancer risk, it is questionable if it deserves to be analyzed together with BRCA1/2. Please discuss.

Please provide literature analysis on controversial role of BRCA1/2 variants in lung cancer risk.

The pathogenic variants deserve to be listed.

Please indicate age, gender and smoking history in pathogenic variant carriers.

There is no real ethnicity-specific variations. Please discuss other relevant findings in other studies.

The analysis of somatic/actionable alterations does not add to the paper.

Round 2

Reviewer 1 Report

Comments and Suggestions for Authors

No other comments

Reviewer 2 Report

Comments and Suggestions for Authors

The authors addressed in an excellent way all issues, which were raised in response to the initial submission